# Virtual Vibrational Analytics of Reduced Graphene Oxide

**DOI:** 10.3390/ijms23136978

**Published:** 2022-06-23

**Authors:** Elena F. Sheka, Nadezhda A. Popova

**Affiliations:** Institute of Physical Researches and Technology, Peoples’ Friendship University of Russia (RUDN University), 117198 Moscow, Russia; nad.3785@mail.ru

**Keywords:** virtual vibrational spectrometry, digital twins, digital spectrometer, unrestricted Hartree–Fock theory, IR and Raman spectra, general frequency kit, virtual vibrational analytics, reduced graphene oxide

## Abstract

The digital twin concept lays the foundation of the virtual vibrational analytics suggested in the current paper. The latter presents extended virtual experiments aimed at determining the specific features of the optical spectra of the studied molecules that provide reliable express analysis of the body spatial structure and chemical content. Reduced graphene oxide was selected as the virtual experiment goal. A set of nanosize necklaced graphene molecules, based on the same graphene domain but differing by the necklace contents, were selected as the relevant DTs. As shown, the Raman spectra signatures contained information concerning the spatial structure of the graphene domains, while the molecule necklaces were responsible for the IR spectra. Suggested sets of general frequency kits facilitate the detailed chemical analysis. Express analysis of a shungite carbon, composed of rGO basic structural units, revealed the high ability of the approach.

## 1. Introduction

It is unlikely that there is at least one person today working in the field of materials science who would not know the widely advertised high-tech material called reduced graphene oxide (rGO). Its “discovery” and a base surge of popularity over the past 10–15 years have been facilitated by the special euphoria of a new direction in materials science-graphenics. Nevertheless, humanity, without realizing it, is familiar with the body from the first bonfire, from the first shovel of coal thrown into the furnace, from the first filler of wheel tires, from the first … and this listing can be continued indefinitely. The ash from a burnt fire, the deposits of natural coal, synthetic black carbon, and, finally, a laboratory product of the chemical reduction of oxidized nanoscale graphite—this is a short list of these unique materials, which are known as *sp*^2^ amorphous carbons, the main structural element of which is a polyatomic corpuscle that might be attributed to rGO. This term refers to a very wide class of new molecular formations called *necklaced graphene molecules* (NGMs), presenting nanometer-sized graphene domains surrounded with heteroatom necklaces of various compositions. The number of corresponding rGO compositions is countless, differing both in the shape and size of the graphene domain and in the content of the heteroatomic necklace. The name ‘necklaced graphene molecules’ is common to all of this richness and better reflects its principle personality than rGO, which was only attributed to a chemically synthesized product.

The graphene era revealed the wide demand for rGO for various applications, which immediately required the formulation and implementation of a reliable analysis of this material, which exists only in solid form. The structure and chemical composition, as a fundamental principle, and the chemical and physical properties as its consequences, became the basis for the development of appropriate analytics. The complexity of the analyzed object and the lack of standard structural and compositional features have inevitably led to the multi-method nature of the required analysis. The diffraction of X-rays and thermal neutron flux, as a source of information about the components of the structure of a solid; thermal gravimetric analysis, as the first cut of the atomic composition in gross as well as in the detection of the presence of hydrogen in the samples; chemical elemental analysis for the presence of carbon, hydrogen, nitrogen, and sulfur atoms; X-ray photoelectron spectroscopic (XPS) analysis for the presence of carbon, oxygen, and atoms of other heavy elements; and energy dispersive spectroscopic analysis of the distribution of these atoms in the mass of a solid all together allowed us to proceed to the stage of creating models of the atomic structure of the sample. Next comes the final analytic stage—verification and approval or rejection of the proposed models. At this stage, the main attention is given to spectral methods of investigation, the primary place among which is occupied by the vibrational spectroscopy of IR absorption and the combinational scattering of light. The latter was discovered independently by Mandelstam and Landsberg in the USSR [1] and Ramanand Krishnan in India [2] and widely known under the name of Raman scattering. The analytic task is evidently highly difficult and not many successful analytic results can be mentioned (see [3,4,5,6,7,8] for a few).

The recent development of virtual vibrational spectrometry (VVS) [9,10,11,12,13] offers an attractive opportunity for significant advancement in this task. Based on a virtual vibrational spectrometer HF Spectrodyn [10], it turned out to be possible to consider nanosized NGMs compatible in the size and composition of heteroatomic necklaces with real basic structural units (BSUs) of *sp*^2^ amorphous carbons [13]. The obtained results allow us to suggest general grounds of *virtual vibrational analytics* (VVA) applied to complicated carbon materials such as rGO. The VVA fundamentals and facilities are considered in the current paper. Digital twins (DTs) and a proper computational code constitute the VVA core. The former, as in the previous cases [13], is presented with a number of NGMs, specified with the same (5, 5) graphene domain, once graphene molecules, but differing by largely variable heteroatom necklaces. The domain is a squared honeycomb sheet with 5 benzenoid units along the armchair and zig-zag edges, respectively. Since hydrogen and oxygen are the main participants that have been revealed in different *sp*^2^ amorphous carbon with experimental analytics [3,4,5,6], only these heteroatoms will be considered in the current study as constituents of necklaces of the relevant DTs, thus related to the hydro-, oxy-, and hydroxy-derivatives of the bare (5, 5) graphene molecule.

The analytic ability of the vibrational spectroscopy of multiatomic molecules, once presented as sets of covalent bonds, is traditionally based on the *general frequency* (GF) concept. The latter is a particular spectral signature of a selected individual covalent bond observed in the corresponding IR and Raman spectra [14,15]. Successfully applied to small molecules, this approach cannot be applied to a case with a number of vibrations of a few hundreds, which is typical to NGMs. To overcome this difficulty, the main goal of the current study was to exhibit not atomically marked individual GFs, but *general frequency kits* (GFKs), attributed to sets of similar covalent bonds. A particular feature of the study concerns a simultaneous consideration of both the density of vibrations and the IR spectra of the relevant DTs, which forms the grounds of the wished spectral marks. A successive result of such an approach is illustrated by the example of a VVA consideration of a shungite carbon.

The rest of this paper is organized as follows. A general description of the VVS technique as well as the formation of DTs are given in Section 2. A brief description of the main unique features of the graphene domain in the Raman spectra is given in Section 3. The first attempt to obtain GFK specific spectral signatures of hydrogen, monoatomic oxygen, hydroxyl, and carboxyl added to the graphene domains as well as of complex oxygen compositions, involving esters, lactones, and acid aldehydes, is considered in Section 4. Discussion of the obtained results and application of suggested GFKs to the analysis of a shungite carbon experimental vibrational spectra is presented in Section 5.

## 2. Grounds of Virtual Vibrational Analytics and the rGO DTs Design

VVS is a chain of successive steps leading to digitalized vibrational analytics according to a general scheme.

### Digital Twins → Virtual Device → IT Product

Here, DTs are the molecular models under study, a virtual device is a carrier of the selected software, and the IT product covers a large set of computational results related to the DTs under different actions in light of the software explored. In the current study, the virtual device HF Specrodyn, implementing the software CLUSTER-Z1 [16], provided the calculation of the harmonic one-quantum spectra of IR absorption and Raman scattering of the DTs of the preliminary optimized structure. The calculations were based on the standard rigid-rotor harmonic-oscillator model in the framework of the semiempirical HF quantum chemical approach. A detailed description of the code is given elsewhere [17]. All of the calculations discussed in the current paper were performed using the AM1 version of the code in the UHF approximation since all of the studied DTs are open-shell molecules [13,18]. It should be noted that such an approach, similar in essence, but less categorically formulated, was practically realized by Yamada’s team previously by applying to both rGO and graphene oxide (GO) [19]. In the framework of the main algorithm of the modern VVS above-mentioned and applying it to the specification, related to different computational tools and suggested previously [12], this approach might be considered as an virtual experiment performed using a DFT Spectrodyn device.

Throughout the paper, the virtual spectra are presented by stick-bars convoluted with a Gaussian bandwidth of 10 cm^−1^ if not pointed otherwise. The intensities are reported in arbitrary units, normalized per maximum values within each spectrum. Since the number of vibrational modes composing the spectra under consideration is too large, the excessive fine structure, statistically suppressed in practice, is covered by trendlines averaged over either 50 or 100 points of linear dispersion of 0.003 ÷ 0.010 cm^−1^ each. The procedure was applied to the plots of the density of vibrations and the Raman scattering spectra.

Unlike the usual attitude toward models for computational spectroscopy, the design of DTs was not subject to the requirement of achieving the best agreement with the experiment. The task of the DT concept is to identify the main including the so far hidden patterns that are characteristic of a given object. The achievement of this result is ensured by the wide variability of the model modifications. Following this trend, two groups of DTs simulating the rGO of different spatial structure and chemical contents were selected for the study. These were NGMs based on the only (5, 5) graphene domain C_66_ (DT **I**) but surrounded with different necklaces. The first group involved compositions with necklaces outside the graphene domain, namely C_66_H_22_ (DT **II**), C_66_O_22_ (DT **III**), C_66_(COOH)_20_ (DT **IV**), and C_66_(OH)_22_ (DT **V**). The former were composed of densely packed monoatomic hydrogen, forming a complete framing of the domain with *sp*^2^C–H methines. The second concerned monoatomic oxygen, providing the domain framing with carbonyls *sp*^2^C=O. The next two DTs were related to necklaces consisting of individual carbonyls *sp*^2^C–COOH and hydroxyls *sp*^2^C–OH. All of the models corresponded to the maximal dense packing. These DTs were supplemented with a similar set of compositions whose necklaces consisted of only four units each, C_66_O_4_ (DT **VI**), C_66_(COOH)_4_ (DT **VII**), and C_66_(OH)_4_ (DT **VIII**), respectively. The second group of DTs resulted from the incorporation of oxygen atoms among the carbon edges of the domain, thus substituting the latter and forming necklaces as groups of esters, lactones, and other compositions, namely, C_60_O_6_ (DT **IX**), C_56_O_10_ (DT **X**), C_70_O_6_ (DT **XI**), C_56_O_14_ (DT **XII**), C_62_O_8_ (DT **XIII**), and C_60_O_12_ (DT **XIV**). Equilibrium structures of the species are presented in Table 1 and Table 2.

## 3. General Frequency Kits of Bare Graphene Domains

A comprehensive VVS analysis of IR absorption and the Raman scattering spectra of bare graphene domains has only recently been conducted. [13]. In light of the analytic focus of the current study, there is a need to briefly repeat its main issues. The first concerns a convincing finding that the IR absorption and Raman scattering of NGMs provide information about their covalent bonds. Thus, the IR spectra present the signatures of the chemical composition of the bonds, laying the foundation of the molecule necklaces, while the Raman spectra of NGMs does the same but are related to the *sp*^2^C–C bond pool of the graphene domains only. The second concerns the dependence of the spectra on structural factors such as the size and shape of the NGMs, on one hand, and the molecule packing, on the other. As it turned out, the IR spectra of the NGMs were practically insensitive to the geometrical factors, but readily responded to changes in the chemical compositions of the necklaces. Conversely, the Raman spectra were highly dependent on the former factor due to the extreme lability of the graphene domain bond structure [13,25], but weakly dependent on the chemical composition of the corresponding necklaces. Accordingly, they were characterized by a rather standard shape, depending only on the geometrical factors, while being similar to the whole community of NGMs. This exclusive feature of the Raman spectra of the NGMs is summarized in Figure 1. Thus, the plot in Figure 1b presents the Raman spectrum of the bare (5, 5) graphene domain (DT **I**) on the background of its density of vibrations (DOV). The domain was a flat sheet that was 1.2 × 1.1 nm^2^ in size. As shown by the DOV trendline, the vibration spectrum was clearly divided into two parts from 20 to 1100 cm^−1^ and from 1100 to 1800 cm^−1^, respectively. This two-part character of the vibration spectrum of NGMs also remained in the crystalline state [21,26,27].

According to the fundamental consideration of the vibrational spectrum of the benzene molecule [20], the first region covered a large set of deformational vibrations involving trigonal *sp*^2^C–C–C bending, benzenoid ring breathing, and *sp*^2^C–C–C puckering (1100–700 cm^−1^), in-plane and out-of-plane *sp*^2^C–C–C bending (700–400 cm^−1^) accompanied with low-frequency breathings of the graphene domain as a whole. In contrast, the region of 1100–1800 cm^−1^ was related to a large set of *sp*^2^C–C stretching. As seen in the figure, Raman scattering mainly involved the latter with a particular favor toward high-frequency ones. These modes also formed in the IR absorption spectrum of the domain [13]. Both spectra had a multiband broad shape. Therewith, the Raman spectra well-followed the DOV trendline, which could be expected for covalent amorphous solids [28]. The similar Raman spectrum was typical to a large set of different NGMs, composed on the basis of the same graphene domain [13].

The Raman spectrum in Figure 1a belongs to a DT related to the synthetic rGO produced in the due course of thermal explosion (TE-rGO) [7,8,13,29]. The latter differs from DT **I** with not only the presence of a complicated necklace, but with a larger size of its graphene domain (9, 9), that is, 2.2 × 2.0 nm^2^. As seen in the figure, the DT Raman spectrum drastically changed, presenting the transformation of a broad multiband structure toward a single-band one. Such a transformation of the Raman spectra was also repeated for the DTs involving bare and hydrogenated (9, 9) graphene domains and became more pronounced in the case of the (11, 11) graphene ones [13]. The matter was connected with a typical size-effect concerning the Raman spectra of substances based on covalent bonds when the substance nature is transformed from an amorphic to a crystalline one [30,31]. In fact, when the linear dimension of the domains achieved and/or exceeded $L_{ph}$~15 nm, which is a free path of the high-frequency optical phonon of the graphene crystal [8,27], the domain dynamics acquired the characteristic features of crystalline behavior and the Raman spectra tended to adopt a single-band appearance in favor of the G-band corresponding to the asymmetrical *sp*^2^C–H stretching.

Aside from the domain size, two more factors come into play. The first is due to the high tendency of graphene domains to stacking. Indeed, the single-layer graphene sheets observed in a number of experiments were only of crystalline structures, the micron linear dimensions of which far exceeded all of the known characteristic dimensional parameters of the graphene crystal quasiparticles, so that their behavior obeys the solid-state laws. Natural packing of the sheets unambiguously led to the sample graphitization. In contrast, in almost all of the *sp*^2^ amorphous carbon structures, the linear dimensions of the domains were smaller than $L_{ph}$, while the domains themselves were packed into stacks that were nanosized in all dimensions. Because of this circumstance, the next important factor for amorphic dynamics comes into play, which is due to the small distance between the layers in stacks.

Indeed, the distance between adjacent layers in graphite-like structures is close to the van der Waals diameter of a carbon atom and constitutes 3.35–3.50 Å [5], thus atoms of neighboring layers touch each other. Nevertheless, the distance between their centers is much larger than a standard size of a single *sp^3^*C–C bond of 1.53 Å, so that the unpaired electrons of carbon atoms in the domain basal planes of neighboring layers cannot be paired. However, as has been shown theoretically [32], a single *sp^3^*C–C bond can be considered broken when the interatomic distance is greater than or equal to 2.5 Å. Later on, the existence of “long” single *sp^3^*C–C bonds was forced to be admitted experimentally [33]. However, the distance between the layers still exceeds the limit value, so that the formation of stable bonds between the layers does not occur and carbon atoms conserve their *sp*^2^ hybridization. At the same time, the vibrations of carbon atoms are not limited in space, which is supported by the occurrence of special out-of-plane phonons in a multilayer graphene crystal, the existence of which significantly changes the form of its Raman spectrum [34,35,36,37]. As noted, these changes are accompanied by the appearance of a clearly seen D band in the vicinity of the G one and the spectrum transformation in the region of 2D bands. Recently, we have proposed the association of a D-band appearance with a group of *dynamic sp^3^*C–C bonds between the domain layers, supported with the displacement of carbon atoms between the layers toward each other [13]. Furthermore, a small distance between the layers may be a favoring factor of an intermode stimulation similar to that of *sp*^2^C–C stretching caused by the presence of oxygen atoms in the molecule carbon core [38]. Additionally, the well-revealed anharmonicity of the out-of-plane displacements, pronouncedly exhibited in the 2D region of the bilayer graphene Raman spectra [34,35,36,37], may lead to the emergence of a broad band on the place of the crystalline D-band in the spectrum of the stack of nanosized graphene domains, as shown in Figure 1c, turning the doublet of the D–G bands of *sp*^2^ amorphous carbons into a sign marking the stack structure of the substance. In fact, the doublet mark of the *sp*^2^ amorphous carbons disappeared when their BSU stacks were completely destroyed [39]. On the other hand, once keeping the D–G doublet image, the full width at the half maximum (FWHM), Δω, and the relative intensities IDIG of the doublet constituents varied quite strongly empirically.

As shown by a detailed analysis of the Raman spectra of an extended set of *sp*^2^ amorphics [8], the former can be attributed to a significant length dispersion, $\Delta l_{C=C}$, related to the *sp*^2^C–C bonds of their graphene domains. In the case of DTs related to the considered NGMs [8,13], the latter constitutes ~0.2 Å, which caused the appearance of the frequency dispersion related to the *sp*^2^C–C stretching Δω≈ 400 cm^−1^ because of the tight interconnection between $\Delta l_{C=C}$ and force constants of the relative vibrations [40,41]. This feature explains why the frequency interval, related to the *sp*^2^C–C stretching in the DOV shown in Figure 1b, is so large. Accordingly, this dispersion is a source of the significant FWHM of both the D- and G-bands. The more ordered the *sp*^2^C–C bond structure, the less Δω, which was empirically confirmed [8].

As for IDIG, intuitively, its variation in practice was first connected with structural imperfections concerning the graphene domains [42,43]. A straight connection of this factor with a linear size of graphene domains was the main topic of the discussion [44,45]. However, a scrupulous analysis of this factor for a set of *sp*^2^ amorphics composed of BSUs with predetermined size [8] did not confirm the suggestion, offering an alternative explanation that connects the ratio magnitude with a particular feature of the graphene domain stacking such as the number of layers, turbostratic packing, and so forth.

Concluding the description of the main characteristics of the vibrational spectra of graphene domains, Table 1 presents the GFKs related to these spectra. The values of the corresponding GFs of the benzene molecule were taken as the reference. Their values, in the case of polyatomic molecules that have been repeatedly confirmed experimentally, were taken as the experimental reference points in the spectra of the graphene domains, which make it possible to evaluate the magnitude of the inevitable blue shifts of the corresponding frequency groups in the virtual spectra [23]. The GFK digits, presented in this and the next table, do not mean individual narrow bands, but are just the center positions of rather broad ones.

## 4. General Frequency Kits of Digital Twins of Reduced Graphene Oxide

We started from the general overlook of the virtual spectra of the NGMs forming the DTs of the first group. Presented in Figure 2 is a set of IR-Raman spectral pairs related to DTs **I**–**V** against the background of their DOVs. The set was opened by the reference spectra related to the bare (5, 5) graphene domain DT **I**. As seen in the figure, the DOV spectra of all of the derivatives strongly retained the common two-part structure, definitely separating the high-frequency stretching from compositions of the deformational modes. In all the spectra, this division occurred at 1100 cm^−1^. The main changes in the DOVs concern the stretching part. A detailed comparison of the DT DOVs with the reference one can be traced in Appendix A.

As seen in Figure 2, all of the Raman spectra resulted from the excitation of the stretching modes and were similar in the shape of broad multiband structures. The difference in the details of their structure clearly shows a change in the pool of *sp*^2^C–C bonds of the pattern graphene domains, caused by the corresponding necklaces, on one hand, as well as the appearance of additional heteroatom covalent bonds, on the other. As for the first fact, it is important to note that even the necklace, consisting of only one hydrogen atom, disturbed the distribution of the *sp*^2^C–C covalent bonds of the domains quite significantly [13]. Despite both facts being connected with the necklaces, thus responding on the chemical composition of the latter, in general, the sensitivity of the Raman spectra to this parameter was not accompanied by pronounced manifestations, as a result, these spectra for different NGMs were more similar than different. As was commented earlier, this is the main fact providing a large similarity of the Raman spectra of the NGMs virtually, and of the *sp*^2^ amorphous carbons in practice [8].

A completely different behavior was typical for the IR spectra. In all cases, the spectra revealed heteroatom covalent bonds characteristic for their necklaces. As seen in Figure 2b, in the case of the fully hydrogenated necklace, deformational *sp*^2^C–H modes provided the spectrum shape. When the necklace included only oxygen atoms (Figure 2c), the light absorption revealed hetero-atomic *sp*^2^C=O stretching. In the case of the mixed hydrogen–oxygen content of the necklaces (Figure 2d,e), both the *sp*^2^C–OH deformations as well as the differently structured *sp*^2^C–O stretching play the role. The shape of all the spectra were highly characteristic, which allowed us to raise the question of the allocation of GFKs. Based on this feature, in what follows, only the IR spectra will be considered to achieve the goal.

### 4.1. Necklaced Graphene Hydrides

Let us begin the consideration of this issue with a detailed analysis of the spectrum of vibrational frequencies using the example of DT **II** related to the necklaced graphene hydride. Figure 3a presents the DT DOV alongside the reference one related to the bare graphene domain DT **I**. According to the fundamental analysis of the dynamics of the benzene molecule [20], hydrogen atoms do not contribute any new vibrational mode in the region of 1100–1800 cm^−1^, attributing the latter to the *sp*^2^C–C stretching only. Nevertheless, as seen in the figure, there was a pronounced change in the DOV spectrum of the considered DT in this region, which exhibited the deformation of the pattern of the graphene domain caused by the derivatization. On the other hand, the changes observed in the low-frequency part can be expectedly connected with the appearance of the *sp*^2^C–H deformational vibrations of different types (see Table 1). In contrast to a previous conclusion [21], the addition of hydrogen atoms on the domain edges deviated the DOV rather significantly. Trendlines plotted in the figure are quite convenient for demonstration of general patterns, but are not very accurate at presenting the detailed difference of the two vibrational spectra. To improve the situation, we transferred to the linear filtration over a lower number of points. Thus, the plots in Figure 3b–d present the differential DOVs ΔV, the trendlines of which were linearly filtered over 50 points.

The plot in Figure 3b presents $\Delta V_{1}=V\left(DT_{curr}\right)-V\left(DT_{ref}\right)$, where $DT_{curr}$ and $DT_{ref}$ mark DTs **II** and **I**, respectively. The obtained $\Delta V_{1}$ reflects both the transformation of the covalent bond pool related to the graphene domains and the introduction of new vibrations related to the necklace atoms. To separate the two parts, the spectrum related to the $DT_{curr}$ carbon core was calculated when removing all necklace atoms of DT **II** and fixing the structure of the remaining carbon atoms. Thus, the obtained $\Delta V_{2}=V\left(DT_{ref}\right)-V\left(Core_{DT_{curr}}\right)$ and $\Delta V_{3}=V\left(DT_{curr}\right)-V\left(Core_{DT_{curr}}\right)$ are plotted in Figure 3c,d, respectively. The plot in Figure 3c clearly reveals the transformation of the vibrational spectrum of the *sp*^2^C–C bonds of the (5, 5) graphene domain caused by the derivatization. As seen in the figure, the hydrogenation concerned all of the vibrational modes. In turn, the plot in Figure 3d is related to the contribution of the necklace atoms to the vibrational spectrum of the $DT_{curr}$. This very $\Delta V_{3}$ lays the foundation for the extraction of the desired GFKs, attributed to the necklaced hydrogenation of the (5, 5) graphene domain. Actually, as seen in Figure 4, $\Delta V_{3}$ correlated well with the IR spectrum of the studied $DT_{curr}$, providing the possibility to suggest the characteristic GFKs listed in Table 1. A structural composition of DT **II** makes it possible to use the reference data in the table belonging to the benzene molecule. As seen in the table, the suggested GFK was well-coherent with the reference data and correlated well with experimental data related to the NGMs, for which the methine-unit content of the necklaces was firmly established [7,24,46].

Obviously, in practice, hydrogeneous necklaces of NGMs are much richer and may involve a variety of hydrogen-containing groups such as *sp^3^*C–H_2_, *sp^3^*C–CH_2_, and *sp^3^*C–CH_3_. The HF Spectrodyn allows for the reliable consideration of all of the cases, based on the (5, 5) graphene domain DTs, as was, say, in the case of *sp^3^*C–CH_3_ [13]. In its turn, fundamental monographs [15,20] offer a highly sophisticated description of GFK related to individual units, which may serve as references. The data showed quite a clear difference related to the different units, which makes them able to distinguish. In this connection, the implementation of the full picture related to the GFKs of the hydrogenated NGMs becomes only a routine computational task. A much more difficult problem awaits us when we move on to oxidized species.

### 4.2. Necklaced Graphene Oxides

#### 4.2.1. Digital Twins of the First Group

Oxygen is the second most frequent element present in carbon materials. Its atoms combine willingly to carbon ones, forming a wide spectrum of organic functionalities from carboxylic acids to ketones and ethers. Figure 1 presents a set of the most prevalent atomic compositions. The latter plays a transcendental role in the optimization of the fabrication and application of highly important materials such as GO and rGO (see [22,47,48] and references therein). Accordingly, the identification of the oxygen-containing units (OCUs) in these materials has become the main analytical goal and IR spectroscopy is at the frontier. However, the practical implementation of the goal has serious problems. The first is due to the fact that standard GFs are related to single individual chemical units in subnanosize molecules while nanoscale chemically modified graphene domains form the BSUs of both GO and rGO and involve a great number of different OCUs in their composition. The second problem concerns the large similarity of GFs [16], which results in unavoidable overlapping values for different OCUs. The next complication arises from the fact that the empirical spectral signatures depend noticeably on the OCUs’ configuration and assembly [14,22,47]. Thus, a successful spectroscopic analysis of the desired material requires a lot of information related to its chemical content and structure. However, even under these conditions, the VVA is quite complicated and is not ambiguous. Based on previous results [13], an evident progress can be expected when substituting GFs with GFKs, which we would like demonstrate below.

As described in Section 2, a particular set of NGM DTs, based on the (5, 5) graphene domain, was selected for the current study. The linear size of the species is compatible with that of BSUs, related to native *sp*^2^ amorphous carbons, which were thoroughly spectroscopically studied [7,8,24]. A general view of the IR and Raman spectra of the first group of DTs **I**–**V** is given in Figure 2. To check the influence of the assembly of the OCUs on the spectra, the necklaces of the DTs **III**–**V** were substantially emptied to four units in each case. The corresponding DTs **VI**, **VII**, and **VIII** are shown in Appendix A. Figure 5 presents a comparable view on the IR spectra of the corresponding DT pairs. To approach the reality, the spectra were broadened, thus providing an opportunity for the reliable fixation of possible GFKs. As seen in the figure, both the difference and similarity in each DT pair are clearly revealed in the spectra. The former demonstrates a remarkable influence of the units assembling of the spectrum shape, thus generalizing the previous conclusion [22] and determining the possible uncertainty in the main band positions of a few tens cm^−1^, in general. At the same time, the observed similarity of the spectra allows us to suggest quite reliable GFKs with the above-mentioned accuracy.

To achieve this goal, we needed reliable reference data similar to those of the benzene molecule discussed in the connection of the GFKs listed in Table 1. Unfortunately, reference data on the spectroscopy of molecules involving heteroatoms other than hydrogen go beyond the benzene molecule. At the same time, the vibrational spectroscopy of polyatomic molecules has accumulated quite a lot of experience in the assigning of covalent bonds including hydrogen and oxygen atoms, in addition to carbon [15,16]. However, seemingly identical bonds manifest themselves differently in the spectra of different molecules, once characterized by a large spread in frequency while covalent bonds, different in structure, are characterized by similar frequencies. This circumstance makes it extremely difficult to choose the reference values of GFK, as a result of which, we turned to the experimental IR spectra of GO and rGO, which have been thoroughly analyzed by now (see [19,22,47,48] and references therein). Thus, the reference part in Table 2 presents the spectral areas, which are the actual ones for the interpretation of the experimental IR spectra of rGO and GO. The areas were supplemented with the GFK assignments, suggested on the basis of the computational vibrational spectroscopy conducted on the particularly selected molecules [22,48], and partially corrected by the authors of the current study.

The calculated data related to DTs **III**–**V** were treated in the manner described previously for the necklaced graphene hydrides in the previous section. The $\Delta V_{3}$ fractions of the DOV spectra are presented in Appendix A alongside the virtual IR spectra of the relevant DTs. Comparing these spectral pairs and taking into account the unavoidable blue shift of the virtual spectral bands [23], it is possible to suggest GFKs related to the corresponding necklaces that are listed in Table 2. Let us consider the obtained results for each DT individually.

The necklace of graphene oxide DT **III** is composed of monoatomic oxygens, the addition of which fully inhibits the dangling bonds of the edge carbon atoms. The circumference composition is extremely similar to that of polycarbonyls (see [49] and references therein) due to which the considered DT may be attributed to graphene polycarbonyl. Analysis of the DOV $\Delta V_{3}$ spectrum, presented in Appendix A, showed that the addition of oxygen atoms causes changes in the molecular spectrum in the large spectral region, which was revealed in the absorption spectrum by the appearance of bands around 750 cm^−1^, 1500 cm^−1^, and 2000 cm^−1^. Distributing the band positions over the reference cells led to the following result. Four groups of vibrational modes, attributed to OCCUs, can be distinguished, namely, out-of-plane *sp*^2^C–O–C bending in the region of 300–1000 cm^−1^ as well as *sp*^2^C–O–C (1200–1600 cm^−1^), *sp*^2^C–C (1500–1600 cm^−1^), and *sp*^2^C=O stretching (1800–1900 cm^−1^). Accordingly, stretching dominated in the IR spectrum, among which the *sp*^2^C–O–C ones covered the largest frequency area, *sp*^2^C–C ones presented carbon atom vibrations stimulated by the presence of chemically added oxygens [22,35], and the *sp*^2^C=O ones were the characteristic signature of carbonyls.

The substitution of oxygen atoms with carboxylic units changed the IR spectrum drastically, as seen when comparing Appendix A. At the same time, distributing the spectrum band positions over the Table 2 cells led to not very much qualitative difference in the GFK assignments of DT **IV** with respect to those of DT **III**. In fact, all four types of the GFK modes previously distinguished still remained while supplemented with the *sp*^2^C–O–H stretching in the DT **IV** case. However, the OCUs were completely different. Based on the oxygen atoms directly attached to the edge atoms of the graphene domain, in the first case, they were related to oxygens of carboxylic units in the second. This caused an expected difference among the modes of the same type. Thus, the out-of-plane *sp*^2^C–O–C bending of DT **III** was transformed in lower-frequency ones related to carboxyls. A large-scale frequency deviation of the *sp*^2^C–O–C stretching of DT **III** remarkably narrowed, while the stimulated *sp*^2^C–C and characteristic *sp*^2^C=O stretching conserved their location. It should be noted that the current experiment revealed that not only oxygen atoms, chemisorbed at edge carbon atoms, but carboxyl units caused the cooperative effect among the functional groups in determining the location and intensity of the IR bands related to the *sp*^2^C–C stretching. Another cooperative effect concerns the presence of additional *sp*^2^C–O–H stretching in the region 2600–3000 cm^−1^. As seen in Figure 5, the relevant band was absent in the spectrum of DT **VII**, apparently, because of the small number of the carboxyl units in the necklace, which were not able to provide a considerable cooperative effect.

A clearly seen, there was considerable reconstruction of the IR spectra of DT **IV** when its necklace carboxylic content became fully hydroxylic (see Appendix A), in fact, which was not so drastic from the GFK viewpoint. As seen in Table 2, filling of the corresponding cells showed that in light of the vibrational mode structure, at first glance, the composition of GFKs related to DT **IV** and DT **V** was very common, despite the relevant OCUs being of different origin. Indeed, low-frequency vibrational modes were well-represented in both cases. However, if in the case of DT **IV** they corresponded to the out-of-plane *sp*^2^C–O–C bending in the space of carboxyl units, *sp*^2^C–OH torsions play a major role supplemented with *sp*^2^C–O–C bending involving oxygen directly attaching to the edge atoms of the domain. For the main pool of the *sp*^2^C–O–C stretching, they are presented in the table with close GFK values despite having different status of oxygens. As in the previous case of DT **III**, directly chemisorbed oxygens of DT V provided a strong stimulation of the *sp*^2^C–C stretching. Comparative analysis was completed by a close similarity of GFKs related to the *sp*^2^C–OH stretching of DT **IV** and DT **V**.

#### 4.2.2. Digital Twins of the Second Group

OCUs, involving oxygen atoms incorporated among edge atoms of the (5, 5) graphene domain, form the basis of the second group of DTs. Their equilibrium structures are presented in Table 2 while the corresponding IR spectra are shown in Figure 6. DT **IX** is related to the case of individual ethers in the molecule circumference. As seen in the table, the DT GFK set expectedly involved *sp*^2^C–O–C stretching, the frequency of which well-correlated with that of the same modes of DTs **III**–**V**. The difference mainly concerns the *sp*^2^C–O–C bending, which should be attributed to the in-plane ones because of the considerable increase in frequency. It should be noted that the absence of the stimulated *sp*^2^C–C stretching showed the lack of cooperative effect promoting the stimulation. The appearance of aggregated ethers in DT **X** did not change the situation in general, leaving the same representatives of the *sp*^2^C–O–C modes while changing the active part of the distribution between the stretching.

Acid anhydrides of DT **XI** provided the simplest IR spectrum, revealing mainly two types of vibrational modes related to the *sp*^2^C–O–C and *sp*^2^C=O stretching. It should be noted that the latter were characterized by 200 cm^−1^ less frequency in comparison with that of the DTs of the first group. This mode feature was repeatedly observed when acid anhydride and lactone OCCUs were present in the molecules [19,22,47,48]. Expectedly, we obtained the same results when considering either the combined composition of aggregated ethers and lactones in DT **XII** or pure lactones, located at the zigzag (DT **XIII**) and armchair (DT **XIV**) edges. All of the discussed peculiarities are presented in Table 2.

## 5. Discussion and Concluding Remarks

In the present work, we abandoned the desire for a virtual substantiation of the experimental spectrum of certain molecules and focused on revealing the characteristic spectral characteristics inherent in the chosen class of molecules. The digital twin concept assists in achieving this goal. The concept lays the foundation of the virtual representations of physical objects or processes along with the connections of data and information that tie the two together [50,51,52]. It can help inspect and conceptualize complex processes, simulate and predict the effects of complex phenomena on the physical counterpart, and interact with or affect the physical counterpart through virtual representation. It has been traditionally associated with domains such as manufacturing and aviation, but has recently been applied to many other diverse areas including science [11]. For the past few years, it has become a frontier issue of virtual vibrational spectrometry [10,11,12,13]. Starting from general grounds related to the DT concept in molecular science, a series of logically consequent steps has brought us to a new vision of the VVS in general, and the power of its application, in particular, thus revealing a new ability of its practical virtual vibrational analytics.

To illustrate the possibilities of VVA, in this work, practical graphenics was chosen as an area that occupies one of the central places in modern materials science. Methodical approaches and the levels of analysis of the obtained results were presented for an important high-tech material such as reduced graphene oxide, thus demonstrating that VVA can already today be attributed to modern analytical methods of a new type.

Since rGO belongs to the class of real *sp*^2^ amorphous carbons, the DTs used in this work were molecular models constructed by taking into account the entire body of knowledge acquired to date concerning these solids. The central place in the information obtained is occupied by a clear idea of the multilevel structure of these solids, which is based on nanosized quasi-molecular formations, or BSUs, which are graphene domains, necklaced with heteroatoms that terminate dangling bonds of the domains edge atoms. The choice of these BSUs as DTs and the possibility of a wide variation in the size and chemical composition of their necklaces is the first fact, which ensures the successful implementation of the VVA. The second element of success is the correct choice of a virtual device that allows for a large number of calculations to be performed at the modern level of the molecular theory of the object under study. In this case, BSUs are stable polyatomic radicals, or molecules with open shells, which determine the use of the UHF version of the HF Spectrodyn as an effective device. A comparative analysis of a large set of calculated data obtained was the completion of the digital cycle. As a result of extensive virtual experiments carried out using two dozen DTs, started in a previous work [13] and completed in the current work, the following conclusions that are important for the VVA of rGO were obtained.

1.The carbon atoms of the graphene domains of the BSUs and heteroatoms of their necklaces participate in the optical vibrational spectra in different ways: the vibrations of the former form the Raman spectra of the BSUs, while the IR spectra are assigned to the vibrations of the heteroatoms of the necklaces.2.The carbon-domain nature of the Raman spectra underlies their unique uniform appearance for a wide range of *sp*^2^ amorphics of various origins. Being multiband in the case of individual BSUs, the Raman spectrum accepts a standard D–G doublet view when packing the BSUs into stacks. In this case, the width of the bands depends on the linear size of the BSUs themselves and the thickness of the stacks. It sharply decreases with an increase in these parameters as their size approaches or exceeds the critical value determined by the mean free path of the optical phonons $L_{ph}$~15 nm of the graphene crystal. The ratio of the band intensities, IDIG is evidently dependent on the degree of ordering of the BSUs’ stack-packing, and the relative increase in the intensity of the D-band may indicate an increase in the thickness of the BSU stacks.3.These conclusions make it possible to carry out an express analysis of the structure of any *sp*^2^ amorphous carbon, accommodating numerous types of rGOs. For example, the typical Raman spectrum of shungite carbon shown in Figure 7a [8] allows us to conclude that the body consisted of BSU stacks, the linear dimensions of whose constituents were less than 15 nm and amounted to several nm. The stack thickness was also less than the specified value, so the stacks consisted of a few BSU layers, which was supported with comparatively equal intensities of the D- and G-bands. A considerable width of both bands allowed us to suspect the turbostratic nature of the packaging. Evidently, the presence of necklaces is one of the reasons for this disordering as well as the slight increase in the interlayer distance.

4.The IR spectra of amorphous *sp*^2^ carbons strongly depend on the chemical content of the necklaces of their BSUs [7]. It has been empirically established that this composition is formed mainly by hydrogen and oxygen atoms, so the DTs considered in this work included only these atoms. The variety of possible compositions of hydrogen- and oxygen-containing units, mainly of the latter, determines the wide variability of empirical IR spectra.5.The DTs considered in the work, which are close in size to empirical BSUs, consist of a hundred atoms, due to which eigenvectors of harmonic vibrations of the molecules involve many hundreds of components. This makes the attribution of the considered vibrational mode to individual atomic pairs, or particular chemical bonds, practically impossible. However, the empirical IR spectra of amorphics retain a certain similarity to the spectra of small molecules, which allows us to speak about the selection of atomic compositions in the realization process of the absorption spectrum. However, if in the case of small molecules such selection leads to the formation of an extensive pool of GFs corresponding to individual covalent bonds, in the present case, we can speak about GFKs, implying the excitation of a set of bonds. In the present work, by changing the composition of the necklaces of the considered DTs in a given way, we succeeded in revealing the GFKs that are related to the composition of the BSU necklaces of different rGOs. A comparison of calculated data with the reference ones revealed a ~250 cm^−1^ blue shift of the former, on average.6.As listed in Table 1 and Table 2, these GFKs allowed us to perform an express analysis of the chemical composition of *sp*^2^ amorphous carbon based on the shape of its IR absorption spectrum. Thus, the IR spectrum of a shungite carbon in Figure 7b showed that the BSUs consisted of a graphene domain of 8–10 nm in size, surrounded with a necklace composed mainly from methine *sp*^2^C–H groups and *sp*^2^C=O carbonyls. There were no hydroxyls in the necklace, while a final conclusion concerning the presence of carboxyl groups, although extremely improbable [20], should be checked by the XRS spectrum.

The given express analysis of the Raman and IR absorption spectra of an rGO sample is only the beginning of the in-depth analysis and is given as an example. Nevertheless, they provide a reliable entry level of analysis, requiring further confirmation using all of the advanced methods of analytical chemistry and spectroscopy [6].

## Data Availability

Any data or material that support the findings of this study can be made available by the corresponding author upon request.

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
