# Peer review of "Virtual Vibrational Analytics of Reduced Graphene Oxide"

_ijms, 2022, doi:10.3390/ijms23136978_

Round 1

Reviewer 1 Report

The authors of the paper which its title is: “Virtual Vibrational Analytix of Reduced Graphene Oxide” evaluated the digital twin concept determining specific features of optical spectra of rGO. The paper is very interesting and well-written with novel finding. I just have couple of comments and questions before publishing:

  • The abstract need to be re0write in order to explain the goals of the paper and bring some results instead of general comments.
  • The introduction part of the paper is well-written; however, I strongly recommend the authors to bring the recent research in the field and then highlight the novelty of their study.
  • In terms of selection of three different configuration for graphene, please explain the reason for choosing them?
  • Please label the peaks of IR in figure 1 to make them more understandable.
  • In the case of mixed hydrogen-307 oxygen content of the necklaces (Figure 2d and 2e), please explain the structure role of deformation more in details.

Reviewer 2 Report

Present article deals with virtual vibrational in rGO. Presented study is important for the scientific community. It is worthy to do research and write article on the mentioned topic. But in the present form the article can not accept. Many statements are written without references.  Article has lots of spelling and English syntax errors. I would like to suggest that authors take help from professional writers.  

Article may accept after addressing the following corrections/comments:

1.     It seems the author is doing self-advertisement. Out of 50, 14 references belong to authors. It is not looking good. I strongly recommend reducing self-citation.

2.       Page-1, Line 27: “from the first ... and this listing can be continued indefinitely.”.:  I think three dots ‘…’ is a typing error. Please rectify it.

3.       Line 20-56: Please provide suitable references for all scientific statements. Follow the same for the entire manuscript.

4.       Line 70-71: “The domain is a 70 squared honeymoon sheet with 5 benzenoid units along armchair and zig-zag edges, 71 respectively.”

What is the meaning of ‘honeymoon sheet’?

5.       Line 154: “In view of analytic foxucing of 154 the current study, there is a need to breafly repeat its main issues.”

Lots of spelling mistakes are there, such as foxucing, breafly etc. Please check for spellings.  

6.       Line 154: What is the significance of NCMs?

7.       Please write the full form of NGM, when it is first time used in manuscript. 

Round 2

Reviewer 2 Report

Author did not correct the manuscript as per previous comment no. 2 and 3. After modification in manuscript as per comment 2 and 3, manuscript can be accepted. Rest answers are satisfactory.
